



# Leading the Lorenz-63 system toward the prescribed regime by model predictive control coupled with data assimilation

Fumitoshi Kawasaki[1], Shunji Kotsuki [2, 3, 4]

[1]Graduate School of Science and Engineering, Chiba University, Chiba, Japan
[2]Institute for Advanced Academic Research, Chiba University, Chiba, Japan
[3]Center for Environmental Remote Sensing, Chiba University, Chiba, Japan
[4]Research Institute of Disaster Medicine, Chiba University, Chiba, Japan

*Correspondence to*: Fumitoshi Kawasaki (fkawasaki@chiba-u.jp), Shunji Kotsuki (shunji.kotsuki@chiba-u.jp)

**Abstract.** In recent years, concerns have been raised regarding the intensification and increase of extreme weather events such as torrential rainfall and typhoons. To mitigate the damage caused by weather-induced disasters, recent studies have started developing weather control technologies to lead the weather to a desirable direction with feasible manipulations. This study proposes introducing the model predictive control (MPC), an advanced control method explored in control engineering, into the framework of the control simulation experiment (CSE). In contrast to previous CSE studies, the proposed method explicitly

considers physical constraints such as the maximum allowable manipulations within the cost function of the MPC. As the first step toward applying the MPC to real weather control, this study performed a series of MPC experiments with the Lorenz-63 model. Our results showed that the Lorenz-63 system can be led to the positive regime with control inputs determined by the MPC. Furthermore, the MPC significantly reduced necessary forecast length compared to earlier CSE studies. It was beneficial to select a member showing a larger regime shift for the initial state when dealing with uncertainty in initial states.

## 1 Introduction

In recent years, concerns have been raised regarding the intensification and increase of extreme weather events such as torrential rainfall and typhoons. To mitigate the damage caused by weather-induced disasters, efforts have been made to improve the forecasting accuracy of stationary heavy rainfall and develop disaster-prevention infrastructures including dams and embankments. Recently, Japan's Moonshot Program started exploring alternative countermeasures for mitigating weather-

induced disasters. Specifically, the program aims at developing weather control technologies to lead the weather to a desirable direction (i.e., regime) with feasible manipulations. Under the program, researchers are exploring various engineering manipulations such as cloud-seeding and atmospheric heating. However, the possible magnitude of human's manipulations for atmosphere is limited. Therefore, simulation studies using numerical weather prediction (NWP) models are needed in addition to the engineering studies to develop effective control approaches with feasible manipulations.





To date, a few simulation studies with NWP models have been conducted for mitigating extreme events. For example, Henderson et al. (2005) conducted numerical experiments using a modified version of the Penn State/NCAR fifth-generation mesoscale model (MM5) 4D-Var to identify the temperature increments required to minimize wind-related damage from Hurricane Andrew in 1992. However, the results may not have sufficient realism due to various experimental limitations (Henderson et al., 2005). The Typhoon Science and Technology Research Center of Yokohama National University proposed using sailing ships and artificial upwelling to reduce the intensity of tropical cyclones. Their simulations demonstrated that the increased drag enhancement by the sailing ships and decreased sea surface temperature by the artificial upwelling successfully weakened tropical cyclones (Fudeyasu et al., 2023; personal communications). The previous studies, however, examined impacts of the manipulations on specific extreme events through control experiments in which simulations with manipulations were simply compared with the simulations without any manipulations. Here, a research framework is necessary to develop effective control approaches with feasible manipulations.

Miyoshi & Sun (2022, hereafter MS22) proposed a control simulation experiment (CSE): an experimental framework for systematically evaluating and exploring control approaches under unknown true values by expanding the observing systems simulation experiment (OSSE). They conducted CSEs with the three-variable Lorenz-63 model (Lorenz, 1963) and succeeded in leading the system to the positive regime with small control inputs. Sun et al. (2023, hereafter SMR23) also applied to CSEs for the 40-variable Lorenz-96 model (Lorenz, 1996), showing that their CSEs succeeded in reducing the number of extreme events of the Lorenz-96 model. Furthermore, Ouyang et al. (2023, hereafter OTK23) successfully reduced the total magnitude of control inputs with Lorenz-63 model by approximately 20 % compared to MS22's approach, by regulating the amplitude of control inputs based on the maximum growth rate of the singular vector. The previous CSE studies (MS22, SMR23, and OTK23) generated control inputs as differences between ensemble members that maintain in and deviate from the desired regime. However, physical constraints, generally needed for real-world applications, cannot be considered explicitly in the previous CSE studies. Therefore, it is worthwhile to explore other methodologies to determine control inputs.

In this study, we propose introducing the model predictive control (MPC) within the framework of CSE. The MPC is an advanced control method that repeats prediction and optimization with explicit consideration of constraints. While the MPC has been widely used in practical fields such as process industry and power electronics (Schwenzer et al., 2021), there has been no study yet that used the MPC for mitigating weather-induced disasters, to our best knowledge. As the first step toward applying the MPC to the real weather control, this study performs a series of MPC experiments with the Lorenz-63 model. Here we explore the way to implement MPC within CSE, and aim to reveal important issues to extend the MPC to high-dimensional NWP models.

The remaining sections of this paper are arranged as follows. Sect. 2 introduces theory of MPC and describes experimental setting. In Sect. 3, we employ a series of MPC experiments with the Lorenz-63 model, and discusses properties of MPC applied to the chaotic dynamical systems. Finally, Sect. 4 provides a summary.





## 2 Method and Experiments

### 2.1 Model predictive control

#### 2.1.1 Definition and procedure

This study explores using MPC for controlling chaotic dynamical system. Here, the MPC is a feedback control method that identifies control inputs to minimize the cost function under constraints at each time. In other word, MPC is a control method that solves an optimal control problem (OCP) for a finite horizon at each time. Strictly speaking, the MPC treated in this study is nonlinear model predictive control (Chen and Shaw, 1982; Keerthi and Gilbert, 1988; Mayne and Michalska, 1990; Mayne et al., 2000).

First, we define the terminology and symbols. As shown in Fig. 1, the two key processes of MPC are model-based prediction and optimization of control inputs in OCP. For these processes, prediction horizon $T_p$ and control horizon $T_c$ are defined independently where subscripts $p$ and $c$ denote prediction and control. Here, $T_p$ ($0 < T_p$) is the length of state prediction, and $T_c$ ($0 < T_c \leq T_p$) is the length of the control inputs to be optimized, respectively. A new axis $\tau$ is the time axis for variables under the optimization, and set to be differently from the time axis $t$. Therefore, $\tau = 0$ denotes the initial times

of the horizons. Furthermore, variables in both horizons are marked with a superscript "$*$"; for example, a state $\mathbf{x}$ at $\tau = \tau_i$ on the horizon at $t = t_i$ is denoted by $\mathbf{x}^*(\tau_i; t_i)$.

   Next, we describe the procedure of the MPC. First, the MPC requires the suitable design of a numerical model $f(\mathbf{x}^*, \mathbf{u}^*)$, a cost function $J(\mathbf{x}^*, \mathbf{u}^*)$, a set of constraints $\mathbf{c}(\mathbf{x}^*, \mathbf{u}^*)$, and a first guess of control inputs $\mathbf{u}^*(\tau; t_i)$ from $\tau = 0$ to $\tau = T_c$ for the desirable control. Now, we consider the process of obtaining control inputs $\mathbf{u}$ at $t = t_i$ based on the MPC.

1.    The present state $\mathbf{x}(t_i)$ is used as the initial state $\mathbf{x}^*(0; t_i)$ for an OCP (i.e., $\mathbf{x}^*(0; t_i) = \mathbf{x}(t_i)$).

   2.    Predicted states $\mathbf{x}^*(\tau; t_i)$ from $\tau = 0$ to $\tau = T_p$ are obtained by the numerical model $f(\mathbf{x}^*, \mathbf{u}^*)$.

   3.    Based on $\mathbf{x}^*(\tau; t_i)$, the solutions $\mathbf{u}^*(\tau; t_i)$ are updated from $\tau = 0$ to $\tau = T_c$ through optimization (cf. Sect. 2.1.2).

   4.    Prediction (step 2) and optimization (step 3) are iterated with updated $\mathbf{u}^*(\tau; t_i)$ and $\mathbf{x}^*(0; t_i)$ until $\mathbf{u}^*(\tau; t_i)$ are sufficiently converged (cf. Sect. 2.1.2).

5.    The control inputs $\mathbf{u}(t)$, taken from finally updated $\mathbf{u}^*(\tau; t_i)$ from $\tau = 0$ to $\tau = k \cdot dt$ ($0 < k \cdot dt < T_c$), are used for the manipulation from $t = t_i$ to $t = t_i + k \cdot dt$ .

   6.    The process returns to step 1 and repeats these processes at $t = t_i + k \cdot dt$.





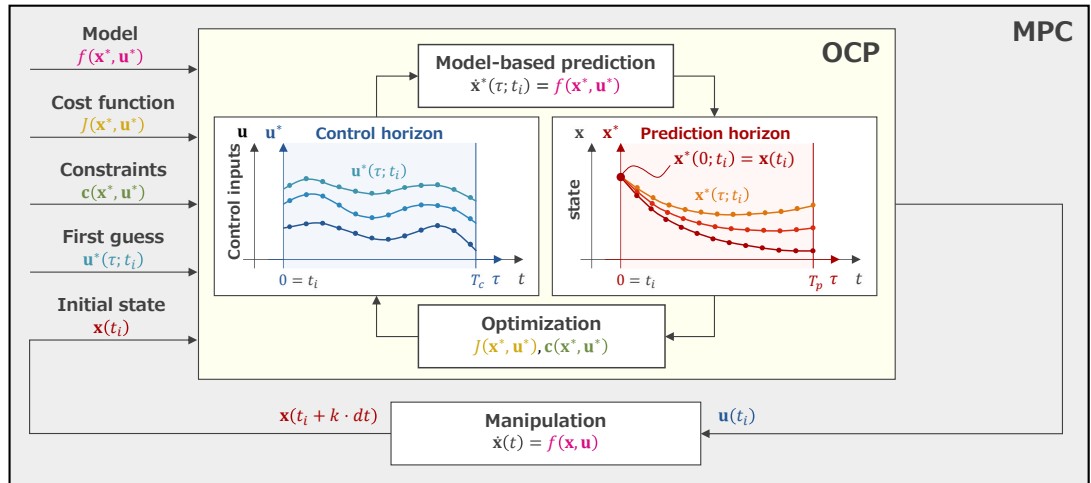

**Figure 1:** Conceptual image of the model predictive control (MPC; the gray block). A numerical model $f(\mathbf{x}^*, \mathbf{u}^*)$, cost function $J(\mathbf{x}^*, \mathbf{u}^*)$, a set of constraints $\mathbf{c}(\mathbf{x}^*, \mathbf{u}^*)$, and a first guess of control inputs $\mathbf{u}^*(\tau; t_i)$ are given to the optimal control problem (OCP; the yellow block). The initial state $\mathbf{x}(t_i)$ is also given by the model time integration with manipulations. The OCP is solved by iterating prediction and optimization until the solutions $\mathbf{u}(t_i)$ are sufficiently converged. Finally, the manipulation is performed by applying the $\mathbf{u}(t_i)$ to $\mathbf{x}(t_i)$. The same process is repeated at the next time ($t = t_i + k \cdot dt$).

### 2.1.2 Optimal control problem

As previously noted, the MPC solves the OCP which identifies control inputs that allow the system to achieve a desirable state for a finite horizon, at each time. Here, we explain that the OCP can be regarded as a variational problem with constraints. We consider a basic OCP with control and prediction horizons being $T = T_c = T_p$ for ease of comprehension. The general equation of state for a nonlinear model and the initial state are given by:

$$\dot{\mathbf{x}}^*(\tau; t) = f\big(\mathbf{x}^*(\tau; t), \mathbf{u}^*(\tau; t)\big), \tag{1}$$

$$\mathbf{x}^*(0; t) = \mathbf{x}(t), \tag{2}$$

where their dimensions are $\mathbf{x}^*(\tau; t) \in \mathbb{R}^n$, and $\mathbf{u}^*(\tau; t) \in \mathbb{R}^l$, respectively. The scalars $n$ and $l$ represent the numbers of model variable and manipulation variables, respectively. The general cost function of the OCP is given by:

$$J(\mathbf{x}^*, \mathbf{u}^*) = \varphi\big(\mathbf{x}^*(T; t)\big) + \int_0^T L\big(\mathbf{x}^*(\tau; t), \mathbf{u}^*(\tau; t)\big)d\tau, \tag{3}$$

where $\varphi\big(\mathbf{x}^*(T; t)\big)$ is the terminal cost, and $L\big(\mathbf{x}^*(\tau; t), \mathbf{u}^*(\tau; t)\big)$ is the stage cost. Both are scalar functions, and various control objectives can be considered by suitable design of these functions. The general constraints of the problem are given by:

$$\mathbf{c}\big(\mathbf{x}^*(\tau; t), \mathbf{u}^*(\tau; t)\big) = \begin{bmatrix} c_1\big(\mathbf{x}^*(\tau; t), \mathbf{u}^*(\tau; t)\big) \\ \vdots \\ c_j\big(\mathbf{x}^*(\tau; t), \mathbf{u}^*(\tau; t)\big) \end{bmatrix} = \mathbf{0}, \tag{4}$$

where $\mathbf{c}\big(\mathbf{x}^*(\tau; t), \mathbf{u}^*(\tau; t)\big) \in \mathbb{R}^j$ is a vector whose elements are equality constraints restricted to zero. The scalar $j$ is the number of constraints. When inequality constraints are imposed, the inequality constraints can be converted to equality





constraints (e.g., by introducing the penalty method or slack variable technique). In summary, the OCP is regarded as the following variational problem that optimizes the cost function, subject to the equation of state and constraints:

$$\text{Minimize: } J(\mathbf{x}^*, \mathbf{u}^*) = \varphi\big(\mathbf{x}^*(T;t)\big) + \int_0^T L\big(\mathbf{x}^*(\tau;t), \mathbf{u}^*(\tau;t)\big)d\tau, \tag{5}$$

$$\text{Subject to: } \begin{cases} f\big(\mathbf{x}^*(\tau;t), \mathbf{u}^*(\tau;t)\big) - \dot{\mathbf{x}}^*(\tau;t) = \mathbf{0} \\ \mathbf{x}^*(0;t) = \mathbf{x}(t) \\ \mathbf{c}\big(\mathbf{x}^*(\tau;t), \mathbf{u}^*(\tau;t)\big) = \mathbf{0} \end{cases}. \tag{6}$$

We note that the equation of state is also regarded as a constraint by transposing $\dot{\mathbf{x}}^*(\tau;t)$ of Eq. (1) to the right-hand side.

The following necessary conditions for optimal control inputs are obtained by converting the constrained problem to an unconstrained problem using the method of Lagrange multipliers (cf. Appendix A):

$$\dot{\mathbf{x}}^*(\tau;t) = f\big(\mathbf{x}^*(\tau;t), \mathbf{u}^*(\tau;t)\big), \tag{7}$$

$$\mathbf{x}^*(0;t) = \mathbf{x}(t), \tag{8}$$

$$\dot{\boldsymbol{\lambda}}^*(\tau;t) = -\frac{\partial H(\mathbf{x}^*, \mathbf{u}^*, \boldsymbol{\lambda}^*, \boldsymbol{\rho}^*)}{\partial \mathbf{x}}, \tag{9}$$


$$\boldsymbol{\lambda}^*(T;t) = \frac{\partial \varphi\big(\mathbf{x}^*(T;t)\big)}{\partial \mathbf{x}}, \tag{10}$$

$$\frac{\partial H(\mathbf{x}^*, \mathbf{u}^*, \boldsymbol{\lambda}^*, \boldsymbol{\rho}^*)}{\partial \mathbf{u}} = \mathbf{0}, \tag{11}$$

$$\mathbf{c}\big(\mathbf{x}^*(\tau;t), \mathbf{u}^*(\tau;t)\big) = \mathbf{0}, \tag{12}$$

where $\boldsymbol{\lambda}^*(\tau, t) \in \mathbb{R}^n$ is the Lagrange multiplier for the equation of state, $\boldsymbol{\rho}^*(\tau;t) \in \mathbb{R}^j$ is the Lagrange multiplier for the constraints, and $H(\mathbf{x}^*, \mathbf{u}^*, \boldsymbol{\lambda}^*, \boldsymbol{\rho}^*)$ is the Hamiltonian defined as follows:

$$H(\mathbf{x}^*, \mathbf{u}^*, \boldsymbol{\lambda}^*, \boldsymbol{\rho}^*) \coloneqq L(\mathbf{x}^*, \mathbf{u}^*) + (\boldsymbol{\lambda}^*)^T f(\mathbf{x}^*, \mathbf{u}^*) + (\boldsymbol{\rho}^*)^T \mathbf{c}(\mathbf{x}^*, \mathbf{u}^*). \tag{13}$$

Derivation of the necessary conditions of the optimal control inputs is detailed in Appendix A. For nonlinear models, it is generally impossible to solve these equations analytically. Thus, this study solve them using a numerical approach. Given the first guess of control inputs $\mathbf{u}^*(\tau;t)$, temporally forward computations (Eqs. 7 and 8) are performed to obtain $\mathbf{x}^*(\tau;t)$ from $\tau = 0$ to $\tau = T$. In this study, zero vectors are selected as the first guess of control inputs $\mathbf{u}^*(\tau;t)$ because the minimization of

control inputs is also included in the cost function $J(\mathbf{x}^*, \mathbf{u}^*)$ as seen later (Eq. 26). Furthermore, $\boldsymbol{\lambda}^*(\tau;t)$ is obtained by temporally backward computations from $\tau = T$ to $\tau = 0$ by Eq. (9) and Eq. (10). Consequently, $\mathbf{u}^*(\tau;t)$ and $\boldsymbol{\rho}^*(\tau;t)$ from $\tau = 0$ to $\tau = T$ can be obtained by applying an optimization algorithm to the nonlinear equations (Eqs. 11 and 12). Therefore, the OCP can be solved by iterating the prediction (Eq. 7) and optimization (Eqs. 11 and 12) until the solutions are sufficiently converged. In this study, the equations (Eqs. 7-12) are discretized with the fourth-order Runge–Kutta scheme and the

Levenberg–Marquardt algorithm is used as the optimization algorithm to solve the nonlinear equations (Eqs. 11 and 12).

When the control horizon is shorter than the prediction horizon ($T_c < T_p$), the necessary conditions for optimal control inputs (Eqs. 7-12) are replaced by:

$$\dot{\mathbf{x}}^*(\tau;t) = \begin{cases} f_c\big(\mathbf{x}^*(\tau;t), \mathbf{u}^*(\tau;t)\big) & (0 \leq \tau < T_c) \\ f_p\big(\mathbf{x}^*(\tau;t)\big) & (T_c \leq \tau < T_p) \end{cases}, \tag{14}$$





$$\mathbf{x}^*(0;t) = \mathbf{x}(t), \tag{15}$$

$$\dot{\boldsymbol{\lambda}}^*(\tau;t) = \begin{cases} -\dfrac{\partial H_c(\mathbf{x}^*,\mathbf{u}^*,\boldsymbol{\lambda}^*,\boldsymbol{\rho}^*)}{\partial \mathbf{x}} & (0 < \tau \leq T_c) \\[2mm] -\dfrac{\partial H_p(\mathbf{x}^*,\boldsymbol{\lambda}^*,\boldsymbol{\rho}^*)}{\partial \mathbf{x}} & \left(T_c < \tau \leq T_p\right) \end{cases}, \tag{16}$$

$$\boldsymbol{\lambda}^*\left(T_p;t\right) = \frac{\partial \varphi\left(\mathbf{x}^*(T_p;t)\right)}{\partial \mathbf{x}}, \tag{17}$$

$$\frac{\partial H_c(\mathbf{x}^*,\mathbf{u}^*,\boldsymbol{\lambda}^*,\boldsymbol{\rho}^*)}{\partial \mathbf{u}} = \mathbf{0} \quad (0 \leq \tau < T_c), \tag{18}$$

$$\begin{cases} \mathbf{c}_c\big(\mathbf{x}^*(\tau;t), \mathbf{u}^*(\tau;t)\big) = \mathbf{0} & (0 \leq \tau < T_c) \\ \mathbf{c}_p\big(\mathbf{x}^*(\tau;t)\big) = \mathbf{0} & \left(T_c \leq \tau < T_p\right) \end{cases}, \tag{19}$$

where the subscript $c$ denotes a function up to the $T_c$ with control inputs $\mathbf{u}^*(\tau;t)$, and the subscript $p$ denotes the function from

$T_c$ to the $T_p$ without control input.

## 2.2 Model predictive control for Lorenz-63 model

### 2.2.1 Lorenz-63 model

This study uses the Lorenz-63 model for MPC experiments. The Lorenz-63 model is a three-variable nonlinear differential equation expressed as follows:

$$\dot{x} = -\sigma x + \sigma y, \tag{20}$$
$$\dot{y} = -xz + rx - y, \tag{21}$$
$$\dot{z} = xy - bz. \tag{22}$$

The model is known to behave in a chaotic manner under certain parameter values. In this study, $\sigma = 10$, $r = 28$, and $b = 8/3$ are selected to form a butterfly pattern with two positive and negative regimes, following previous studies (MS22; OTK23).

Moreover, the model is discretized and integrated using the fourth-order Runge–Kutta scheme. One time step of integration is defined as $dt = 0.01$ unit of time throughout this study. With the Lorenz-63 model, the state vector becomes $\mathbf{x} = [x, y, z]^T$ and the number of model variable is $n = 3$.

### 2.2.3 The optimal control problem with the Lorenz-63 model

This study considers a control problem: to keep the Lorenz-63 system in the positive regime ($x \geq 0$) following the previous

studies (MS22, and OTK23). Note that our approach includes minimization of the control inputs owing to Eq. (27). The equations of state (Eq. 14) are given by:

$$f_c(\mathbf{x}^*, \mathbf{u}^*) = \begin{bmatrix} -\sigma x^* + \sigma y^* + u_x^* \\ -x^* z^* + r x^* - y^* + u_y^* \\ x^* y^* - b z^* + u_z^* \end{bmatrix}, \tag{23}$$





$$f_p(\mathbf{x}^*) = \begin{bmatrix} -\sigma x^* + \sigma y^* \\ -x^* z^* + r x^* - y^* \\ x^* y^* - b z^* \end{bmatrix}, \tag{24}$$

where $\mathbf{x}^* = [x^*, y^*, z^*]^T$, $\mathbf{u}^* = [u_x^*, u_y^*, u_z^*]^T$. As previously noted, one of the control objectives in this problem is leading the Lorenz-63 system to the positive regime. Therefore, the inequality constraint $x^*(\tau; t) \geq 0$ is imposed from $\tau = 0$ to $\tau = T_p$. In this study, the penalty method is introduced to treat the inequality constraint, and the penalty function for $x^*(\tau; t) \geq 0$ is defined as follows:

$$P_{x^* \geq 0}(x^*) := \frac{1}{2}\{\max(-x^*, 0)\}^2. \tag{25}$$

The inequality constraint can be considered in the cost function as follows. Including the minimization of the control inputs, the cost function is given by:

$$J = \int_0^{T_c} \left\{ \frac{1}{2}(\mathbf{u}^*)^T \mathbf{u}^* + \alpha_{x^* \geq 0} \cdot P_{x^* \geq 0}(x^*) \right\} d\tau + \int_{T_c}^{T_p} \alpha_{x^* \geq 0} \cdot P_{x^* \geq 0}(x^*) d\tau + \alpha_{x^* \geq 0} \cdot P_{x^* \geq 0}\left(x^*(T_p; t)\right), \tag{26}$$

where $\alpha_{x^* \geq 0} > 0$ is the tunable penalty parameter that balances weights of magnitude of control inputs $(\frac{1}{2}(\mathbf{u}^*)^T \mathbf{u}^*)$ and the inequality constraint $(x^*(\tau; t) \geq 0)$ in the cost function. The third term of Eq. (26) corresponds to the terminal cost, and is necessary for considering explicitly terminal condition of $\mathbf{x}^*(\tau; t)$. This study employs $\alpha_{x^* \geq 0} = 10^4$ from our preliminary investigations. Consequently, the necessary conditions for optimal control inputs (Eqs. 14-19) are formulated to following equations for the control problem of the Lorenz-63 model:

$$\dot{\mathbf{x}}^*(\tau; t) = \begin{cases} f_c(\mathbf{x}^*, \mathbf{u}^*) = \begin{bmatrix} -\sigma x^* + \sigma y^* + u_x^* \\ -x^* z^* + r x^* - y^* + u_y^* \\ x^* y^* - b z^* + u_z^* \end{bmatrix} & (0 \leq \tau < T_c) \\ f_p(\mathbf{x}^*) = \begin{bmatrix} -\sigma x^* + \sigma y^* \\ -x^* z^* + r x^* - y^* \\ x^* y^* - b z^* \end{bmatrix} & (T_c \leq \tau < T_p) \end{cases}, \tag{27}$$

$$\mathbf{x}^*(0; t) = \mathbf{x}(t), \tag{28}$$

$$\dot{\boldsymbol{\lambda}}^*(\tau; t) = -\begin{bmatrix} -\lambda_x^* \sigma + \lambda_y^*(-z^* + r) + \lambda_z^* y^* - \alpha_{x^* \geq 0} \cdot \max(-x^*, 0) \\ \lambda_x^* \sigma - \lambda_y^* + \lambda_z^* x^* \\ -\lambda_y^* x^* - \lambda_z^* b \end{bmatrix} \quad (0 < \tau \leq T_p), \tag{29}$$

$$\boldsymbol{\lambda}^*(T_p; t) = \begin{bmatrix} -\alpha_{x^* \geq 0} \cdot \max(-x^*, 0) \\ 0 \\ 0 \end{bmatrix}, \tag{30}$$

$$\frac{\partial H_c(\mathbf{x}^*, \mathbf{u}^*, \boldsymbol{\lambda}^*, \boldsymbol{\rho}^*)}{\partial \mathbf{u}} = \begin{bmatrix} u_x^* + \lambda_x^* \\ u_y^* + \lambda_y^* \\ u_z^* + \lambda_z^* \end{bmatrix} = \mathbf{0} \quad (0 \leq \tau < T_c), \tag{31}$$

Where $\boldsymbol{\lambda}^* = [\lambda_x^*, \lambda_y^*, \lambda_z^*]^T$. As discussed later, this control problem can be extended to other experimental settings such as manipulating only one-variable control input (cf. Sect. 3.3) and adding a constraint for a $L^2$ norm of control inputs (cf. Sec. 3.4).





## 2.3 Control simulation experiment with model predictive control

The CSE is an experimental framework that controls nature run (NR), extended from OSSE. The key concept of CSE is that the true state of the NR is unknown but manipulations can be added to the NR, assuming a realistic atmosphere.

Based on previous studies (Kalnay et al., 2007; Yang et al., 2012; MS22; OTK23), the experimental setting of our CSE is determined as follows. We first employed a free run with the Lorenz-63 model for 2,009,000 steps without any manipulations. The initial values of the free run are generated by random numbers $\mathcal{N}(0.0, 2.0)$ for $x$, $y$, and $z$ independently. Observations are generated at every $T_o = 8$ steps by adding uncorrelated Gaussian noise $\varepsilon \sim \mathcal{N}(0.0, 2.0)$ into the free run where the subscript $o$ denotes the observation. The DA cycles are performed by assimilating the generated observations for the last 2,008,000 steps by an Ensemble Kalman Filter (EnKF) (Evensen, 1994). This study employs the perturbed observation method (Burgers et al., 1998) as the EnKF to obtain a stable analysis ensemble under the nonlinear system (Lawson and Hansen, 2004). We discarded the first 8,000 steps from of the 2,008,000-step DA cycle for CSE. The root-mean-square errors (RMSEs) and multiplicative inflation parameters of two-million DA cycle are shown in Table 1. In this study, 1,000 independent CSEs for 2,000 steps are performed from different starting points to evaluates the CSEs statistically. OTK23 noted that starting points around the large $x$ are generally difficult for leading the system to the positive regime for the Lorenz-63 model. Therefore, the 1,000 different starting point are sampled sequentially from the points satisfying $0 \leq x < 15$ in the two-million-step DA cycle. We employ three indicators to evaluate CSEs. The first index is the success rate (SR) which denotes the percentage of cases that satisfy $x \geq 0$ for entire experimental period (i.e., 2,000 steps) among the 1,000 CSEs. The mean total failure (MTF) and mean total control inputs (MTC) are defined as the mean of $\sum_{x<0} x \cdot dt$ and $\sum \|\mathbf{u}\| \cdot dt$ of the 1,000 CSEs, respectively.

The procedure of CSE with MPC is designed as follows:

1. At a certain time $t = t_i$, the observation $\mathbf{y}^o(t_i)$ is simulated from the NR.

2. DA is employed to obtain an analysis ensemble $\mathbf{X}^a(t_i)$.

3. The ensemble forecast $\mathbf{X}^b(t)$ from $t = t_i$ to $t = t_i + T_p$ is computed from the analysis ensemble $\mathbf{X}^a(t_i)$.

4. If at least one member indicates a regime shift (RS) during the ensemble forecast, the process continues to step 5. Otherwise, the NR is evolved until $t = t_i + T_o$ and returns to step 1.

5. The OCP is solved to obtain control inputs $\mathbf{u}(t)$ from $t = t_i$ to $t = t_i + T_o$ from control inputs after iterations $\mathbf{u}^*(\tau; t_i)$ from $\tau = 0$ to $\tau = T_o$.

6. The NR is evolved from $t = t_i$ to $t = t_i + T_o$ by applying the obtained control inputs $\mathbf{u}(t)$. In addition, the $\mathbf{X}^b(t)$ from $t = t_i$ to $t = t_i + T_o$ is computed by applied the same control inputs to the analysis ensemble $\mathbf{X}^a(t_i)$ for DA at the next time. Notably, the control inputs are applied to $\dot{\mathbf{x}}(t)$ through the numerical model $f(\mathbf{x}, \mathbf{u})$ (cf. Eq. 1), rather than direct addition to $\mathbf{x}(t)$.

7. The process returns to step 1 and repeats these processes at $t = t_i + T_o$.

Here, $\mathbf{X} \in \mathbb{R}^{n \times m}$ is an ensemble of state and $m$ is the ensemble size. Superscripts $a$ and $b$ denote analysis and background, respectively.





This procedure is illustrated in Fig. 2. For simplicity, the flow diagrams of the CSE are divided into two cases: without a RS in Fig. 2 (a) and with a RS in Fig. 2 (b). The procedure of CSE for forecasts without a RS in Fig. 2 (a) is identical to the

OSSE. In contrast, the procedure of CSE for forecasts with a RS in Fig. 2 (b) has additional processes for identifying and applying control inputs. The upper panel of Fig. 2 (c) shows a conceptual image of identifying control inputs, and the lower panel shows an application of control inputs to the NR through the Lorenz-63 model. Importantly, the NR cannot be used as the initial state of the OCP because it is always unknown. Therefore, an analysis ensemble is used as the initial state. As discussed later (Sect. 3.5), the initial state for OCP substantially affects the control results, and the member with the smallest

state $x$ (i.e., the largest RS) in the ensemble forecast (step 3) is selected as the initial state in this study unless otherwise specified. In addition, $T_c = 8$ steps are selected throughout this study from our preliminary investigations.

**Table 1:** The RMSEs and the multiplicative inflation parameters used in this study for each ensemble size $m$. The multiplicative inflation is applied to background ensemble perturbations. The inflation parameters were manually tuned so that analysis RMSEs are minimized over

the two-million-step OSSEs.

| Ensemble size: $m$ | 10 | 20 | 30 | 40 | 50 | 100 |
|---|---|---|---|---|---|---|
| RMSE | 0.393 | 0.300 | 0.282 | 0.277 | 0.273 | 0.271 |
| Inflation | 1.50 | 1.18 | 1.08 | 1.06 | 1.04 | 1.02 |



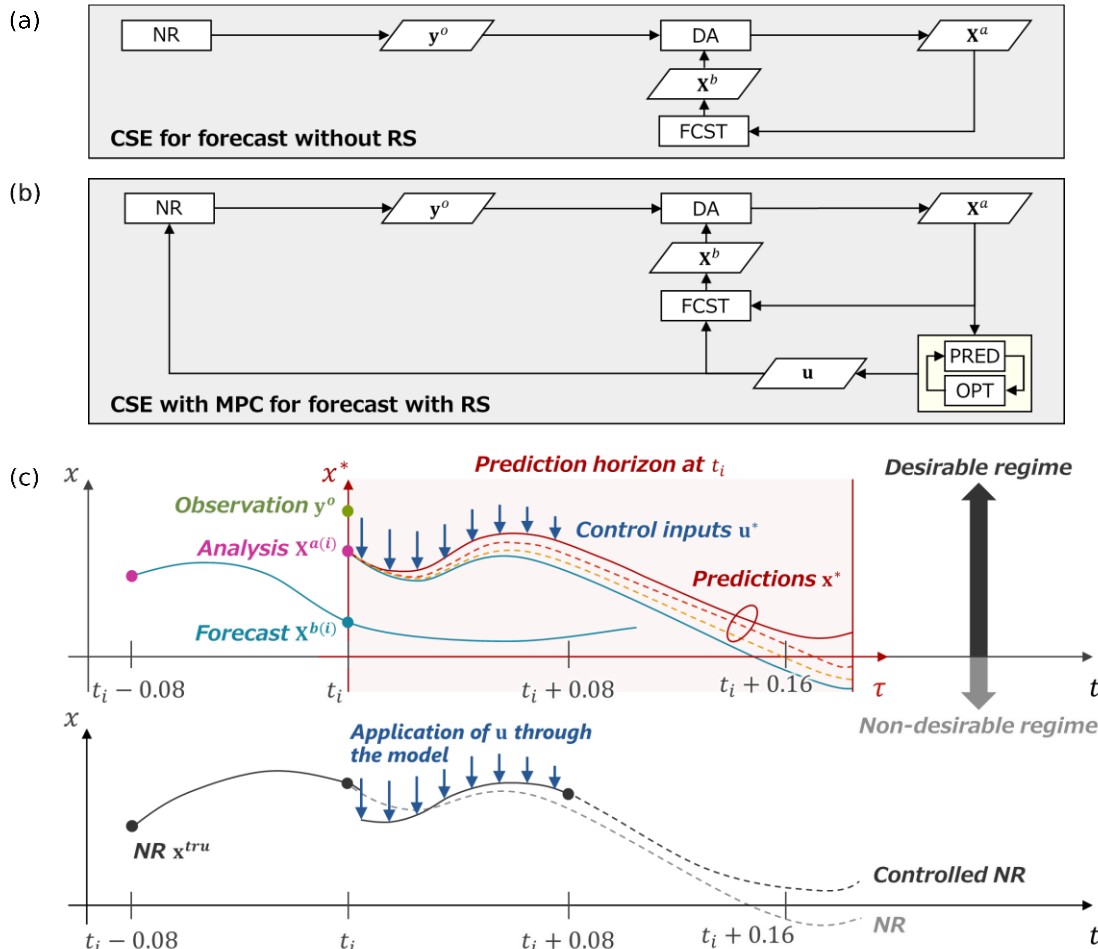

**Figure 2:** Flow diagram and conceptual image of CSE with MPC. (a) Flow diagram of CSE for forecasts without a RS, which is identical to OSSE. (b) Flow diagram of CSE with MPC for forecasts with a RS, which has additional processes for identifying and applying control inputs. (c) Conceptual image of CSE with MPC. The upper panel shows an image of identifying control inputs, and the lower panel shows an application of control inputs to the NR.





# 3 Results and Discussion

## 3.1 Impacts on the nature run

First, CSE is conducted with the Lorenz-63 model to verify the impacts of MPC on the NR. The control objective is leading the system to the positive regime under minimization of the three-variable control inputs. Here, $T_p = 20$ steps and $m = 50$ are selected as discussed later in Sect. 3.2.

   The NR and the $L^2$ norm of control inputs $\|\mathbf{u}\|$ are shown in Fig. 3 and Fig. 4. The butterfly pattern appears in Fig. 3 (a) because no control input is applied. In contrast, the NR successfully keeps the positive regime with consideration of the

inequality constraint $x^* \geq 0$ by the MPC in Fig. 3 (b) and Fig 4 (a). This result indicates that the NR can be controlled by the short forecast(i.e., $T_p = 20$) steps. Importantly, the unit of $\|\mathbf{u}\|$ is identical to $\dot{\mathbf{x}}$, rather than $\mathbf{x}$. Therefore, the magnitude of the control inputs added during $dt = 0.01$ are $\|\mathbf{u}\| \cdot dt$. As demonstrated in Fig. 4 (b), the maximum value of magnitude of the control inputs added during $dt$ is approximately $40 \cdot 0.01 = 0.4$, which is smaller than the maximum value of state.

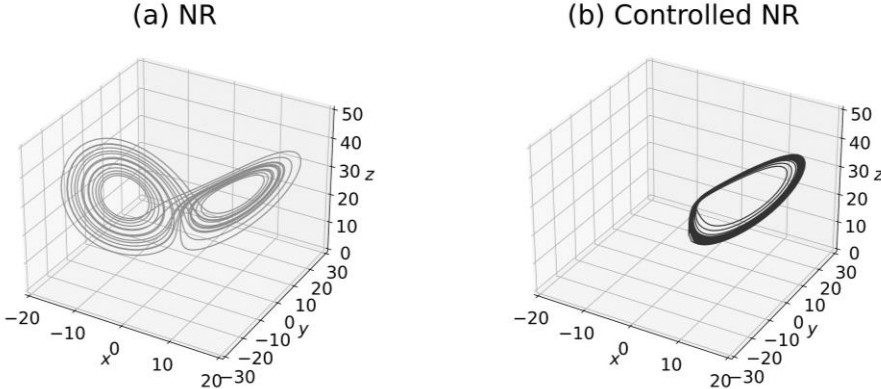


**Figure 3:** The NR and controlled NR of the Lorenz-63 model for 2,000 steps. Each starting point is selected from the 24th step of two-million DA cycle. (a) shows the uncontrolled NR without MPC. (b) shows the controlled NR by the MPC with $T_p = 20$ steps, $T_c = 8$ steps, and $m = 50$.


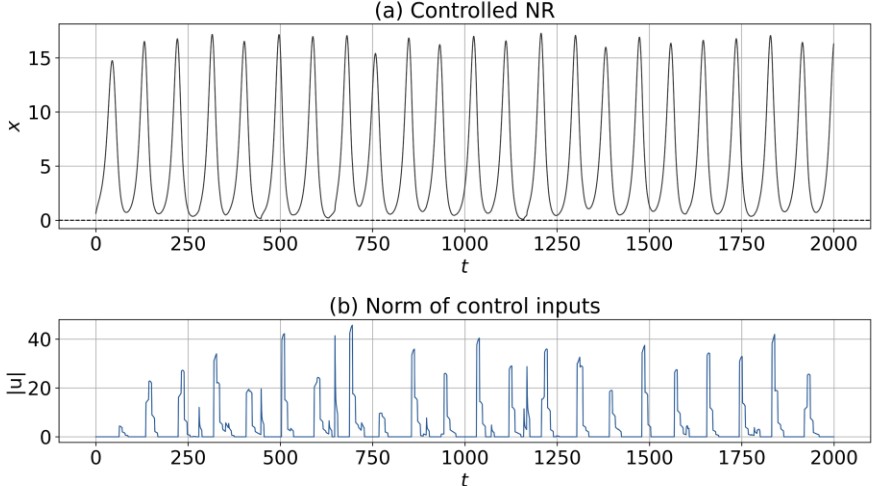

**Figure 4:** The controlled NR and the $L^2$ norm of control inputs with $T_p = 20$ steps, $T_c = 8$ steps, and $m = 50$. The starting point is the 24th step of two-million DA cycle. (a) shows the time series of state $x$. (b) shows the $L^2$ norm of control inputs $\|\mathbf{u}\|$.

Figure 5 shows the prediction of the state and optimization of the control inputs in each horizon at an arbitrary selected step (the 232nd step of the MPC experiment of Fig. 4). Since the forecast (blue dotted line) from the initial state shows a RS, the control is activated to solve the OCP. As demonstrated in Fig. 5 (a), the trajectory of the controlled prediction gradually shifts to satisfy $x^* \geq 0$ by iterative computations; finally, $x^* \geq 0$ is satisfied (red solid line). The uncontrolled NR shows a RS (gray dotted line); in contrast, the controlled NR can avoid the RS (black solid line) through the addition of control inputs (Fig. 5 (b), (c), and (d)) after iterations. Note that the final prediction in OCP and the controlled NR are not identical because the prediction in OCP used an initial state from the member with the largest RS, rather than the NR.

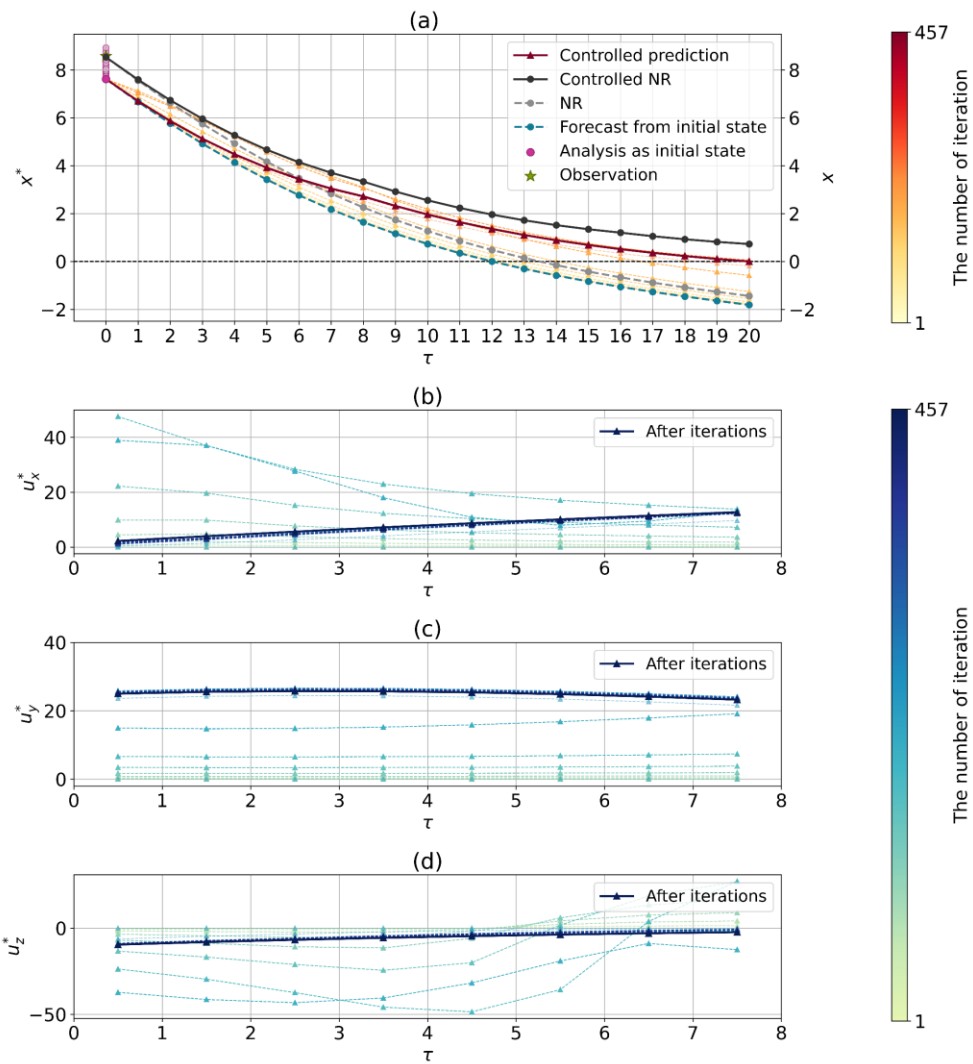

**Figure 5:** The prediction of the state and optimization of the control inputs in the horizon at an arbitrary selected step (the 232nd step of the MPC experiment of Fig. 4). Iterative computations were performed 457 times for solving the OCP in this case. (a) The predictions, NRs, forecast, analysis, and observation in $T_p$. Panels (b), (c) and (d) show the control inputs $u_x^*$, $u_y^*$ and $u_z^*$ during optimization in $T_c$.

### 3.2 Sensitivity to the prediction horizon and the ensemble size

Here, we investigate the sensitivity to $T_p$ and $m$ for MPC performance. For that purpose, we conducted 1,000 independent CSEs and summarized their SR, MTF, and MTC in Fig. 6. The darker color in Fig. 6 indicates better controllability. Higher values of $m$ generally yield better results, increasing the SR and reducing MTF and MTC. However, improvements owing to the increased ensemble size $m$ converge for $m \geq 50$ in many cases. The reasons for the improved results with larger ensemble





size $m$ will be discussed in Sect. 3.5. In addition, the results with shorter $T_p$, such as $T_p = 10$ steps, tend to be worse; especially

the MTC would increase because the control would be difficult by delaying the timing of control activation. On the other hand, longer $T_p$ would not necessarily improve the results. In particular, it considerably worsens at $T_p = 50$ steps, presumably because of linear approximation errors involving state evolution in $T_p$.

It should be noted that a higher SR does not necessarily indicate less MTF. For example, focusing on $m = 30$, the SR of $T_p = 10$ steps (SR $= 0.487$) is much lower than the SR of $T_p = 40$ steps (SR $= 0.921$). However, the MTF of $T_p = 10$

steps (MTF $= -7.9 \times 10^{-3}$) is less than the MTF of $T_p = 40$ steps (MTF $= -2.7 \times 10^{-2}$). Therefore, control would fail more frequently, but not significantly, with $T_p = 10$ steps than with $T_p = 40$ steps.

Hereafter, the experiment with $T_p = 20$ steps and $m = 50$ are considered to be a standard experimental setting in this study because the parameters yielded one of the best performances. The SR, MTF, and MTC in several experimental settings with $T_p = 20$ steps and $m = 50$, including the experiments discussed later (cf. Sect. 3.3 and Sect. 3.4), are summarized in

Table 2.

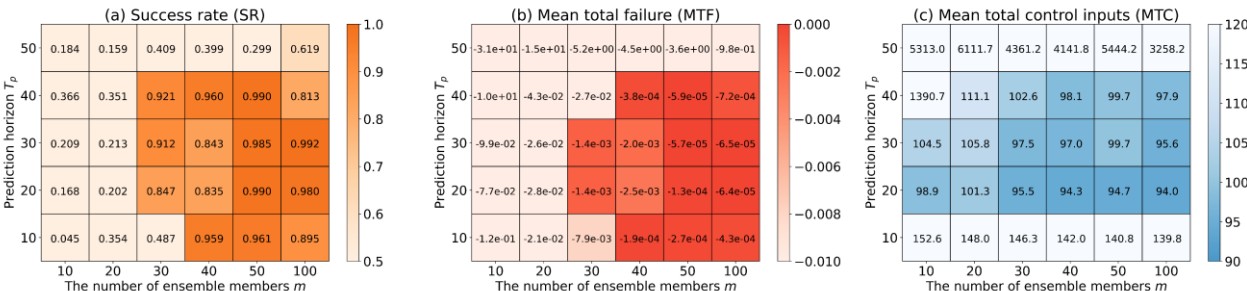

**Figure 6:** Sensitivity to the prediction horizon $T_p$ and the ensemble size $m$ with three evaluation indicators: (a) success rate (SR), (b) mean total failure (MTF), and (c) mean total control inputs (MTC). Darker colors in (a), (b) and (c) indicate better

controllability.





**Table 2:** Summary of success rate (SR), mean total failure (MTF) and mean total control inputs (MTC) for results in each experimental setting with $T_p = 20$ steps and $m = 50$. (a) shows the results of standard MPC experiment. (b) shows the results of CSEs with only one-variable control input, and (c) shows the results of CSE with an additional constraint for the $L^2$ norm of control inputs.

| | Manipulation variable | Constraints | Success rate (SR) | Mean total failure (MTF) | Mean total control inputs (MTC) | Section |
|---|---|---|---|---|---|---|
| (a) | $u_x^*, u_y^*, u_z^*$ | $x^* \geq 0$ | 0.990 | $-1.34 \times 10^{-4}$ | 94.7 | Sect. 3.2 |
| (b) | $u_x^*$ | $x^* \geq 0$ | 0.789 | $-4.85 \times 10^{-3}$ | 358.2 | |
| | $u_y^*$ | $x^* \geq 0$ | 0.956 | $-1.18 \times 10^{-4}$ | 132.3 | Sect. 3.3 |
| | $u_z^*$ | $x^* \geq 0$ | 0.021 | $-3.41$ | 1403.2 | |
| (c) | $u_x^*, u_y^*, u_z^*$ | $x^* \geq 0, \|\mathbf{u}^*\| \leq 20$ | 0.932 | $-9.19 \times 10^{-4}$ | 111.2 | |
| | | $x^* \geq 0, \|\mathbf{u}^*\| \leq 30$ | 0.959 | $-5.15 \times 10^{-4}$ | 126.6 | Sect. 3.4 |
| | | $x^* \geq 0, \|\mathbf{u}^*\| \leq 40$ | 0.980 | $-2.22 \times 10^{-4}$ | 131.9 | |

### 3.3 MPC experiments with one-variable control input

For realistic control scenarios, it is important to consider control problems in which limited control inputs relative to model dimensions are available. Here, this section investigates the CSE with one-variable control input. Figure 7 (a), (b), and (c) show the NRs controlled by only $u_x$, $u_y$, or $u_z$, respectively. While the NR controlled by $u_x$ (Fig. 7 a) shows a pattern fluctuating around $x = 0$, the NR controlled by $u_y$ (Fig. 7 b) exhibits a pattern similar to the case of three-variable control inputs (Fig. 3 b). Intriguingly, the NR controlled by $u_z$ demonstrates a random pattern that does not significantly deviate from $x \geq 0$. In addition, the SR, MTF, and MTC for the 1,000 CSEs are listed in Table 2 (b). Compared with the case of three-variable control inputs presented in Table 2 (a), the case with only $u_y$ is slightly inferior yet comparable; the controllability of case with $u_x$ is more difficult, and the difficulty escalates further when employing $u_z$.


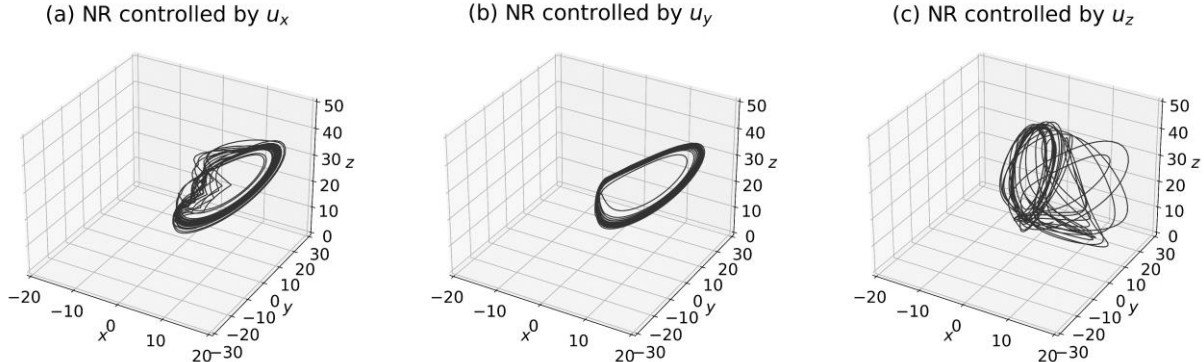

**Figure 7:** NRs controlled by one-variable control input: (a) controlled by $u_x$, (b) controlled by $u_y$, and (c) controlled by $u_z$. Each starting point is the identical to Fig. 3 (i.e., the 24th step of two-million DA cycle).

### 3.4 MPC experiment constrained by magnitudes of control inputs

Here, we show that the MPC can consider constraints for control inputs in addition to the constraint for state (i.e., $x^* \geq 0$). Therefore, we consider MPC experiments with additional inequality constraints: $L^2$ norms of the control inputs $\|\mathbf{u}^*\| \leq U$ ($U = 20, 30, 40$). Namely, this section discusses MPC experiments constrained by magnitudes of control inputs. For that purpose, the penalty method is also introduced to treat $\|\mathbf{u}^*\| \leq U$, which is given by:

$$P_{\|\mathbf{u}^*\| \leq U}(\|\mathbf{u}^*\|) := \frac{1}{2}\{\max(\|\mathbf{u}^*\| - U, 0)\}^2. \tag{32}$$

In this study, $\alpha_{\|\mathbf{u}^*\| \leq U} = 10^3$ is selected as the penalty parameter for $\|\mathbf{u}^*\| \leq U$ from our preliminary experiments.

Figure 8 shows the NRs and the $L^2$ norm of the control inputs with additional $\|\mathbf{u}^*\| \leq U$. In all cases of $U$, NRs (Figs. 8 a, c and e) indicate patterns similar to the case without $\|\mathbf{u}^*\| \leq U$ (Fig. 3 b). The $L^2$ norm of the control inputs $\|\mathbf{u}^*\|$ satisfies the constraint for each $U$ (Figs. 8 b, d, and f), especially for larger $U$. However, with a smaller $U$ (i.e., $U = 20$), the $L^2$ norm of control inputs occasionally exceeds the prescribed upper limit significantly. This is because the penalty method adds a penalty weighted by $\alpha_{\|\mathbf{u}^*\| \leq U}$ to the cost function, and does not guarantee to satisfy the constraint every time. Therefore, different results can be obtained by adjusting $\alpha_{\|\mathbf{u}^*\|}$. For example, by increasing $\alpha_{\|\mathbf{u}^*\|}$, $\|\mathbf{u}^*\| \leq U$ can be more strictly satisfied instead of decreasing the weight for $x^* \geq 0$. Their SR, MTF, and MTC for the 1,000 CSEs are presented in Table 2 (c). Compared with the result in the absence of $\|\mathbf{u}^*\| \leq U$ listed in Table 2 (a), the result with $\|\mathbf{u}^*\| \leq U$ is worse overall because the constraint impose more difficulty on the control problem. In addition, the MTC decrease for smaller $U$, but the SR and MTF worsen accordingly.



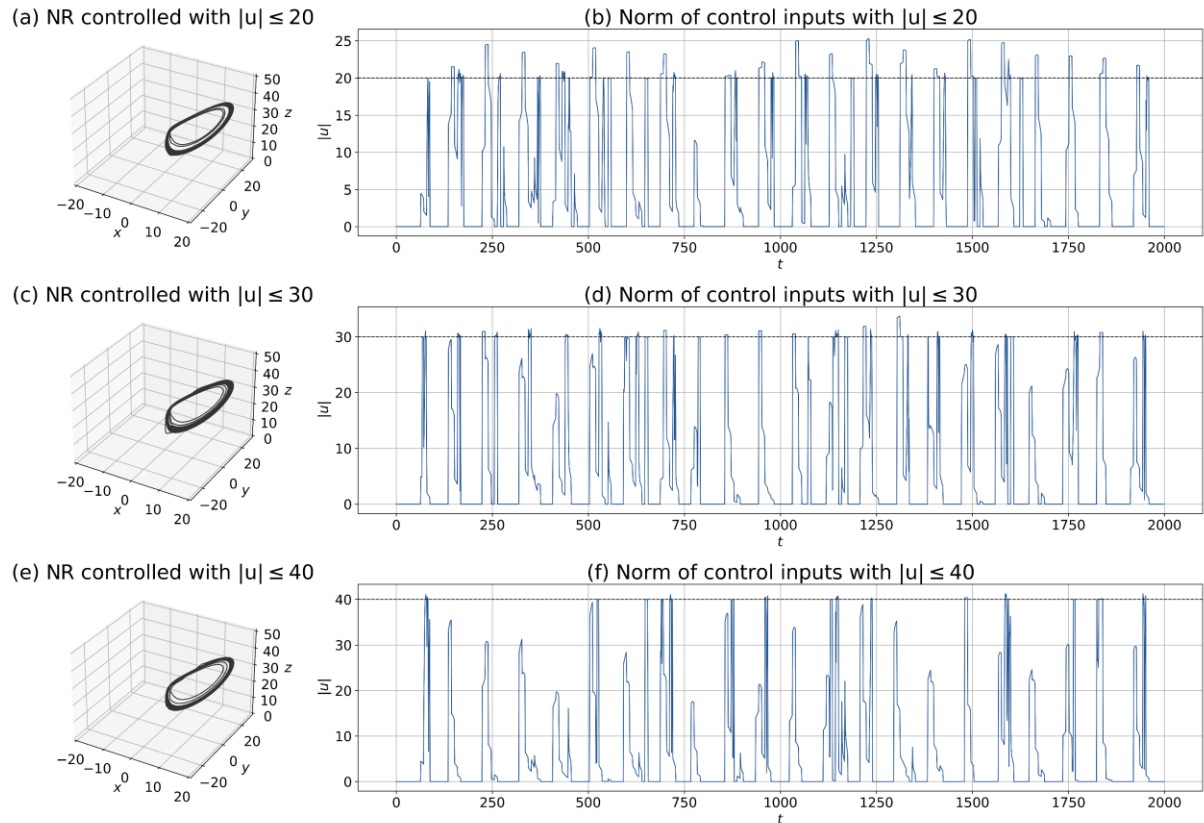

**Figure 8:** The MPC experiments with inequality constraints for control inputs $\|\mathbf{u}^*\| \leq U$: (a, b) $U = 20$, (c, d) $U = 30$, and (e, f) $U = 40$. (a, c, e) show the NRs, and (b, d, f) show the $L^2$ norm of control inputs. The dashed lines in (b, d, f) show the prescribed upper limits of the control inputs (i.e., $U = 20, 30$ and $40$). Each starting point is identical to Fig. 3 (i.e., the 24th step of two-million DA cycle).

**3.5 Sensitivity to the initial state**

For controlling NR, it would be preferable to use the NR as the initial state for identifying control inputs. However, the state estimated by DA must be used because the true value is always unknown. Therefore, there is uncertainty in MPC-derived control inputs based on the estimated states by DA. This uncertainty may not cause serious problems for some systems without strong nonlinearity. Chaotic dynamical systems, however, require careful explorations on options for stable control because small uncertainties can cause large differences. Here, we discuss the initial state that would be valid for leading a chaotic dynamical system to a prescribed regime.

We performed 1,000 independent CSEs and computed SR, MTF, and MTC for five kinds of initial states: "Random (all mem.)", "Mean (all mem.)", "Random (RS mem.)", "Mean (RS mem.)", and "Largest (RS mem.)", respectively. The "(all

mem.)" label denotes selection from among all members in the analysis ensemble, and the "(RS mem.)" label denotes selection from among the members of the analysis ensemble showing RSs. The "Random" label denotes a randomly sampled member,

the "Mean" label denotes the mean of the members, and the "Largest" label denotes the member showing the largest RS. For example, "Mean (all mem.)" indicates mean analysis ensemble. The results are shown in Fig. 9. The experiment of "Largest (RS mem.)" yielded the best results, showing the highest SR, and the smallest MTF and MTC. Furthermore, Fig. 9 shows that it was better to use a member selected from the "(RS mem.)", rather than "(all mem.)", as the initial state. We presume that it is safer to select a member showing a larger RS for the initial state when uncertainty exists in initial state. Therefore, the

improvement with a larger ensemble size $m$ in Sect. 3.2 is attributed to the fact that larger ensemble size $m$ can provides a member with a larger RS. Consequently, obtaining a member with a larger RS would be important for successfully leading chaotic dynamical systems to the prescribed regime by MPC.

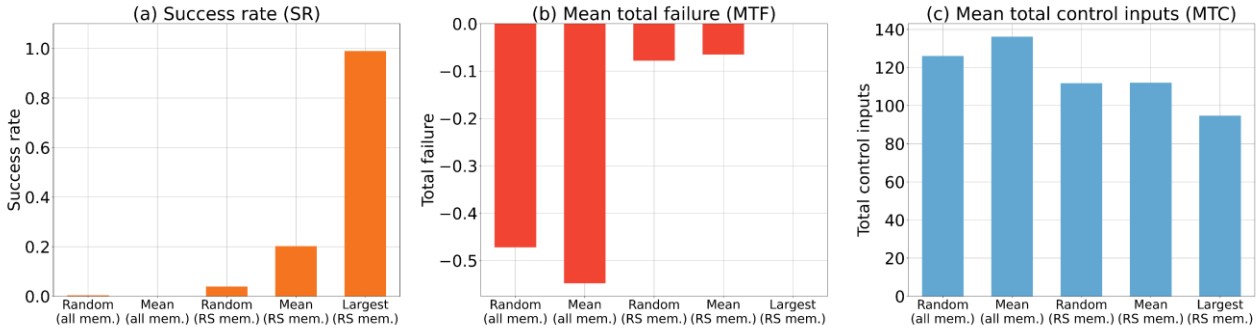

**Figure 9:** Sensitivity to the initial state with $T_p = 20$ steps and $m = 50$. Three evaluation indicators are shown for (a) success rate (SR), (b) mean total failure (MTF), and (c) mean total control inputs (MTC), respectively. The "(all mem.)" label denotes selection from among all members in the analysis ensemble, and the "(RS mem.)" label denotes selection from among the members of the analysis ensemble showing RSs. The "Random" label denotes a randomly sampled member, the "Mean" label denotes the mean of the members, and the "Largest" label denotes the member showing the largest RS.

**4 Conclusions**

In this study, we propose introducing the MPC within the framework of CSE. The advantage of using MPC is that control objectives and constraints can be explicitly considered. Therefore, we expect that this approach will be useful for realistic weather control by designing a cost function and constraints suitably.

     We conducted MPC experiments with the Lorenz-63 model and successfully led the system to the positive regime. The

previous CSE studies (MS22 and OTK23) required longer forecasts (about 300 steps) for successful controls with the Lorenz-63 model, whereas our approach required much shorter forecasts such as 20 steps. We also confirmed that controllability would





be difficult with limited variables of control inputs or with additional constraints. In our discussion, we suggested that it is safer to select a member showing a larger RS for the initial state when dealing with uncertainty in initial states.

This study is an investigation of the first phase of MPC for weather control. In the future, this approach will be investigated with more realistic NWP models. In addition, several improvements remain for the MPC to be applied to weather control. Our present approach requires many iterations to solve the OCP, and temporally forward and backward computations are required for each iteration. Therefore, it is computationally difficult to apply the present approach to large-dimensional NWP models as it is. Therefore, further studies are needed to explore faster approaches to solve OCPs for high-dimensional models.

Finally, we emphasize caution in weather control research. The achievement of control for extreme events would be an innovative way to mitigate weather-induced disasters. However, the side effects of weather control must be carefully examined from an ethical, legal, and social issues (ELSI) perspective. Our research program also addresses such social issues with legal and ethical researchers. Further ELSI research will be also conducted to satisfy responsible and innovative research for the weather control studies.

**Appendix A: Derivation of the necessary conditions for optimal control inputs**

Here, we derive the necessary conditions for optimal control inputs. For simplicity, we consider the following problem:

$$\text{Minimize: } J(\mathbf{x}^*, \mathbf{u}^*) = \varphi\big(\mathbf{x}^*(T;t)\big) + \int_0^T L\big(\mathbf{x}^*(\tau;t), \mathbf{u}^*(\tau;t)\big)d\tau, \tag{A1}$$

$$\text{Subject to: } \begin{cases} f\big(\mathbf{x}^*(\tau;t), \mathbf{u}^*(\tau;t)\big) - \dot{\mathbf{x}}^* = \mathbf{0} \\ \mathbf{x}^*(0;t) = \mathbf{x}(t) \\ \mathbf{c}\big(\mathbf{x}^*(\tau;t), \mathbf{u}^*(\tau;t)\big) = \mathbf{0} \end{cases}. \tag{A2}$$

A Lagrangian is introduced to convert the constrained problem to an unconstrained problem. The Lagrangian is defined as:

$$\tilde{J}(\mathbf{x}^*, \dot{\mathbf{x}}^*, \mathbf{u}^*, \boldsymbol{\lambda}^*, \boldsymbol{\rho}^*) \coloneqq J(\mathbf{x}^*, \mathbf{u}^*) + \int_0^T [(\boldsymbol{\lambda}^*)^T\{f(\mathbf{x}^*, \mathbf{u}^*) - \dot{\mathbf{x}}^*\} + (\boldsymbol{\rho}^*)^T \mathbf{c}(\mathbf{x}^*, \mathbf{u}^*)]d\tau. \tag{A3}$$

In addition, a Hamiltonian is defined as follows:

$$H(\mathbf{x}^*, \mathbf{u}^*, \boldsymbol{\lambda}^*, \boldsymbol{\rho}^*) \coloneqq L(\mathbf{x}^*, \mathbf{u}^*) + (\boldsymbol{\lambda}^*)^T f(\mathbf{x}^*, \mathbf{u}^*) + (\boldsymbol{\rho}^*)^T \mathbf{c}(\mathbf{x}^*, \mathbf{u}^*). \tag{A4}$$

Then, $\tilde{J}$ is represented using $H$; it is divided into $\dot{\mathbf{x}}^*$ terms and other terms in the integral as follows:

$$\tilde{J}(\mathbf{x}^*, \dot{\mathbf{x}}^*, \mathbf{u}^*, \boldsymbol{\lambda}^*, \boldsymbol{\rho}^*) = \varphi\big(\mathbf{x}^*(T;t)\big) + \int_0^T \{H(\mathbf{x}^*, \mathbf{u}^*, \boldsymbol{\lambda}^*, \boldsymbol{\rho}^*) - (\boldsymbol{\lambda}^*)^T \dot{\mathbf{x}}^*\}d\tau. \tag{A5}$$

The stationary condition of $\tilde{J}$, which does not have constraints explicitly, is equal to the stationary condition of the original constrained problem. Namely, the original constrained problem was converted to an unconstrained problem. We note that this is not valid for special cases in which the linear independence constraint qualification is not satisfied. The stationary condition of $\tilde{J}$ is that its variation $\delta\tilde{J}$ (i.e., infinitesimal change) is zero. Applying Taylor expansion and disregarding higher than second-order terms of $\delta\mathbf{x}^*$ and $\delta\mathbf{u}^*$, $\delta\tilde{J}$ is given by:

$$\delta\tilde{J} = \tilde{J}(\mathbf{x}^* + \delta\mathbf{x}^*, \dot{\mathbf{x}}^* + \delta\dot{\mathbf{x}}^*, \mathbf{u}^* + \delta\mathbf{u}^*, \boldsymbol{\lambda}^*, \boldsymbol{\rho}^*) - \tilde{J}(\mathbf{x}^*, \dot{\mathbf{x}}^*, \mathbf{u}^*, \boldsymbol{\lambda}^*, \boldsymbol{\rho}^*)$$
$$= \left(\frac{\partial \varphi(\mathbf{x}^*(T;t))}{\partial \mathbf{x}}\right)^T \delta \mathbf{x}^*(T;t) + \int_0^T \left\{\left(\frac{\partial H(\mathbf{x}^*,\mathbf{u}^*,\lambda^*,\rho^*)}{\partial \mathbf{x}}\right)^T \delta \mathbf{x}^* + \left(\frac{\partial H(\mathbf{x}^*,\mathbf{u}^*,\lambda^*,\rho^*)}{\partial \mathbf{u}}\right)^T \delta \mathbf{u}^* - (\lambda^*)^T \delta \dot{\mathbf{x}}^* \right\} d\tau$$

$$= \left\{\left(\frac{\partial \varphi(\mathbf{x}^*(T;t))}{\partial \mathbf{x}}\right)^T - (\lambda^*(T;t))^T\right\} \delta \mathbf{x}^*(T;t) + (\lambda^*(0;t))^T \delta \mathbf{x}^*(0;t)$$

$$+ \int_0^T \left\{\left(\left(\frac{\partial H(\mathbf{x}^*,\mathbf{u}^*,\lambda^*,\rho^*)}{\partial \mathbf{x}}\right)^T + (\dot{\lambda}^*)^T\right) \delta \mathbf{x}^* + \left(\frac{\partial H(\mathbf{x}^*,\mathbf{u}^*,\lambda^*,\rho^*)}{\partial \mathbf{u}}\right)^T \delta \mathbf{u}^*\right\} d\tau. \tag{A6}$$

Importantly, $\delta \mathbf{x}^*(0;t) = 0$ because $\mathbf{x}^*(0;t)$ is fixed by $\mathbf{x}(t)$. In addition, $\delta \lambda^*$ and $\delta \rho^*$ are disregarded because consideration

of these variations only yields conditions already obtained (i.e., $f(\mathbf{x}^*(\tau;t), \mathbf{u}^*(\tau;t)) - \dot{\mathbf{x}}^* = \mathbf{0}$ and $\mathbf{c}(\mathbf{x}^*(\tau;t), \mathbf{u}^*(\tau;t)) = \mathbf{0}$).

According to Eq. (A6), the condition for $\delta \tilde{J}$ to be zero is that the coefficients of $\delta \mathbf{x}^*$ and $\delta \mathbf{u}^*$ are zero. By summarizing the

conditions from Eq. (A6), the equation of state, and the other constraints, the necessary conditions for optimal control inputs

can be derived as follows:

$$\dot{\mathbf{x}}^*(\tau;t) = f(\mathbf{x}^*(\tau;t), \mathbf{u}^*(\tau;t)), \tag{A7}$$

$$\mathbf{x}^*(0;t) = \mathbf{x}(t), \tag{A8}$$

$$\dot{\lambda}^*(\tau;t) = -\frac{\partial H(\mathbf{x}^*,\mathbf{u}^*,\lambda^*,\rho^*)}{\partial \mathbf{x}}, \tag{A9}$$

$$\lambda^*(T;t) = \frac{\partial \varphi(\mathbf{x}^*(T;t))}{\partial \mathbf{x}}, \tag{A10}$$

$$\frac{\partial H(\mathbf{x}^*,\mathbf{u}^*,\lambda^*,\rho^*)}{\partial \mathbf{u}} = \mathbf{0}, \tag{A11}$$

$$\mathbf{c}(\mathbf{x}^*(\tau;t), \mathbf{u}^*(\tau;t)) = \mathbf{0}. \tag{A12}$$


*Code availability.* The code that supports the findings of this study is available from the corresponding author upon reasonable request.

*Data availability.* The authors declare that all data supporting the findings of this study are available within the figures and tables of the paper.

*Author contribution.* FK and SK conceptualized this study. FK conducted the numerical experiments and wrote the manuscript. SK supervised and directed this study.


*Competing interests.* The authors declare that they have no conflict of interest.

*Acknowledgements.* This study was partly supported by the JST Moonshot R&D (JPMJMS2284, JPMJMS2389), the Japan Society for the Promotion of Science (JSPS) KAKENHI grants JP21H04571, and the IAAR Research Support Program of

Chiba University. The authors thank program members of the Moonshot JPMJMS2284 for valuable discussion.



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
