# Peer review of "Leading the Lorenz-63 system toward the prescribed regime by model predictive control coupled with data assimilation"

_Nonlinear Processes in Geophysics, 2024_

## Referee Comment (RC2)

Review of the paper:

**_Leading the Lorenz-63 system toward the prescribed regime by model predictive control coupled with data assimilation_**
*by Fumitoshi Kawasaki and Shunji Kotsuki*

**1. General comments**

The paper discusses a Control Simulation Experiment (CSE) framework for evaluating and exploring control approaches in weather forecasting. It involves manipulating Nature Run (NR) of numerical models together with model predictive control (MPC) in order to lead the dynamical system to prescribed regimes of the states. Previous studies, well described as a review provided in the introduction by the authors, already showcased the potential of this experimental framework. In the idea, the CSE framework enables control of the NR with feasible manipulations assuming a realistic atmosphere, thus aiming to develop effective control approaches for extreme weather events for instance.

The paper is overall well written and comes with a set of Figures that helps the reader to understand how MPC is used in the context of CSE studies. The procedures and equations are correctly presented. The concepts are tested on the Lorenz 63 system with successful results. The experiment is particularly well detailed with many evaluations of different setups and according analysis.

I recommend publication after minor review taking into account some additional questions (2) and correction of typing errors (3).

**2. Specific comments:**

**Q.1: l.197.** The authors mentioned that "*OTK23 noted that starting points around the large* x *are generally difficult for leading the system to the positive regime for the Lorenz-63 model*".
- Though intuitive, is there any additional reason to explain this result in the corresponding study?
- Also, would it be possible to improve this by the method proposed in the paper? It seems important because this means that the approach would improve the capability to drive the system back to prescribed regimes from "extreme states", which is one of the main objectives of such approaches as described by the authors in the abstract.

**Q.2: l.275.** Would it be possible to add a Figure presenting on the Y-axis the three metrics w.r.t the length of state prediction on the X-axis. That would help to read the comments made by the authors from l.274 to 277.

**Q.3: l.303.** This random pattern is also not consistent with the butterfly wing of the positive L63 regime. Can you explain a bit more why, according to your opinion?

**Q.4: Conclusion, l.372.** The authors said: "it is computationally difficult to apply the present approach to large-dimensional NWP models as it is. Therefore, further studies are needed to explore faster approaches to solve OCPs for high-dimensional models". This relates to potential applications of the research findings in realistic weather control and designing cost-effective strategies for mitigating extreme events.

- Can the authors provide some lines of research with appropriate citations to look for solutions in high-dimensional dynamical systems? For instance, looking at a smaller representation of the system with projection on latent space? Maybe also looking at fast solvers for the optimization of the OCP?
- regarding the ethical considerations, I would also mention that mitigating extreme events may also lead to shift the entire dynamical regime of the system in high-dimensional space, with no extreme events but with other unseen/unknown characteristics that may not be beneficial on other aspects (for the biodiversity, the wind/sun-related power production for instance).

**Q.5: Data availability.** Would it be possible to make the code open source, as a Git repository with code and Notebooks for instance? Indeed the EnKF L63 experiments used in third work is often used by the community and it would be nice to make available both:

- the DA-L63 setup,
- together with the MPC code presented in the paper, to ensure reproducibility of the results and provide a quickstart initial setup for future works and people interested in collaborating on this topic.

**3. Technical corrections**

Please find below a list of grammatical or typing errors to consider before publication:

**l.20-22**: The authors used exactly the same 2 first sentences for the abstract and Introduction. This has to be modified.

**l.26**: In the sentence "Under the program, researchers are exploring various engineering manipulations such as cloud-seeding and atmospheric heating," the word "manipulations" may be replaced with "techniques" for a more precise and formal tone. Or at least, well define what you intend by "manipulations".

**l.127**: The word "solve" in the sentence "Thus, this study solve them using a numerical approach" should be changed to "solves."

**l.128**: In the sentence "Given the first guess of control inputs $\mathbf{u}_*(\tau; t)$, temporally forward computations (Eqs. 7 and 8) are performed to obtain $\mathbf{x}_*(\tau; t)$ from $\tau = 0$ to $\tau = T$," the term "temporally" should be changed to "temporal" for accuracy.

**l.366:** The phrase "the constraint impose more difficulty" should be corrected to "the constraint imposes more difficulty."

**l.326:** In the sentence "In addition, the MTC decrease for smaller $U$," the verb should be in singular form as "decreases" to match the subject "MTC." In the same sentence, "but the SR and MTF worsen accordingly" could be improved by adding "do" before "worsen" for better clarity.

---

## Author Comment (AC1)

**[Response to Reviewer's Comments]**

We thank the reviewers for her/his careful reviews and for kindly giving us valuable and constructive comments and suggestions that helped us improve our manuscript. Here, we provide our point-by-point responses, indicated in blue. This PDF file would be useful to check the revised manuscript. The L in point-by-point responses corresponds to the line of the revised manuscript. Changes in the revised manuscript are indicated in red.
* * *
**[Reviewer 1 General Comments]**

The manuscript put the ideas that have been suggested by Miyoshi and Sun (2022) into a more robust optimal control formulation, although there is no reference to optimal control theory but their terminology is used. The optimal control technique is applied to different control situations as well as different components of the Lorenz 1963 model.

This is a well presented manuscript that introduces some interesting possibilities for weather control, and II only have minor questions that need addressing for clarity.

Response: We are grateful for your constructive feedback. We revised the manuscript following comments.

**[Reviewer 1 Specific Comments]**

(1) I may have missed it but define what slack variables are in paragraph 110.

Response: Slack variables are not defined in the manuscript because this study does not use the slack variable technique. We added more clarification on this point (L109).

(2) Same paragraph, the sentence after the equations refers to a right hand side but of which equation in the set?

Response: The sentence in L117 indicates that transposing $\dot{\mathbf{x}}^*(\tau; t)$ in Eq. (1) to the right-hand side results in the first equation of Eq. (6). We emphasized this point in the revised manuscript (L117).

(3) Paragraph 135: Define what is meant by the Lavenberg-Marquardt algorithm and why is it important that you use this one here?

Response: We added the description of the algorithm (L138).

(4) Lines 173 and 175: Again I may have missed this but what do you mean by terminal cost and conditions?

Response: We added the descriptions on this point (L179). We note that "terminal condition" was

replaced by "terminal state" to make its meaning clearer.

(5) A comment rather than a question but well done for stating on line 222 that the NR cannot be used as the initial state as we do knot know what the true state is.
Response: We appreciate your positive feedback.

(6) A remark/question: In the conclusions you mention weather control but we have to take into account latency, in that in the time it has taken the algorithm to converge the atmospheric state may have changes significantly enough from the prediction that the control is null and void. Just something to keep in mind.
Response: We fully agree with your opinion. We added the description on this point (L382).

**[Reviewer 2 General Comments]**

The paper discusses a Control Simulation Experiment (CSE) framework for evaluating and exploring control approaches in weather forecasting. It involves manipulating Nature Run (NR) of numerical models together with model predictive control (MPC) in order to lead the dynamical system to prescribed regimes of the states. Previous studies, well described as a review provided in the introduction by the authors, already showcased the potential of this experimental framework. In the idea, the CSE framework enables control of the NR with feasible manipulations assuming a realistic atmosphere, thus aiming to develop effective control approaches for extreme weather events for instance.

The paper is overall well written and comes with a set of Figures that helps the reader to understand how MPC is used in the context of CSE studies. The procedures and equations are correctly presented. The concepts are tested on the Lorenz 63 system with successful results. The experiment is particularly well detailed with many evaluations of different setups and according analysis.

I recommend publication after minor review taking into account some additional questions (2) and correction of typing errors (3).

Response: We appreciate your positive and careful comments. We revised the manuscript following suggestions.

**[Reviewer 2 Specific Comments]**

(1) l.197. The authors mentioned that "OTK23 noted that starting points around the large $x$ are generally difficult for leading the system to the positive regime for the Lorenz-63 model".

● Though intuitive, is there any additional reason to explain this result in the corresponding study?
Response: OTK23 has shown that starting points around large $x$ are difficult to leading the system to the positive regime in their experiments, but unfortunately no additional information was described. However, the behavior of the Lorenz-63 system shows that the amplitude of $x$ generally increases before the

regime shifts. Therefore, it is likely to occur a regime shift in a point with large $x$.

- Also, would it be possible to improve this by the method proposed in the paper? It seems important because this means that the approach would improve the capability to drive the system back to prescribed regimes from "extreme states", which is one of the main objectives of such approaches as described by the authors in the abstract.

Response: We added the discussion on this point (L421, Appendix B).

(2) l.275. Would it be possible to add a Figure presenting on the Y-axis the three metrics w.r.t the length of state prediction on the X-axis. That would help to read the comments made by the authors from l.274 to 277.

Response: We carefully considered this suggestion. We believe that it is possible to understand the corresponding comments from Fig. 6, and adding further figures with similar messages would be redundant. Consequently, we decided not to add the figure. Thank you for your suggestion.

(3) l.303. This random pattern is also not consistent with the butterfly wing of the positive L63 regime. Can you explain a bit more why, according to your opinion.

Response: We added the discussion on this point (L311). Regarding Fig. 6 (c), a bug was found in the visualization code, and the behavior has changed accordingly. The revised results indicate the unstable pattern as well as the before revision, therefore, the essential characteristics did not change due to this bug fix. We are very sorry about this.

(4) l.372. The authors said: "it is computationally difficult to apply the present approach to large-dimensional NWP models as it is. Therefore, further studies are needed to explore faster approaches to solve OCPs for high-dimensional models". This relates to potential applications of the research findings in realistic weather control and designing cost-effective strategies for mitigating extreme events.

- Can the authors provide some lines of research with appropriate citations to look for solutions in high-dimensional dynamical systems? For instance, looking at a smaller representation of the system with projection on latent space? Maybe also looking at fast solvers for the optimization of the OCP?

Response: We added the discussion and citations on this point (L381).

- regarding the ethical considerations, I would also mention that mitigating extreme events may also lead to shift the entire dynamical regime of the system in high-dimensional space, with no extreme events but with other unseen/unknown characteristics that may not be beneficial on other aspects (for the biodiversity, the wind/sun-related power production for instance).

Response: We totally agree with your opinion. We added the description on this point (L387).

(5) Data availability. Would it be possible to make the code open source, as a Git repository with code and

Notebooks for instance? Indeed the EnKF L63 experiments used in third work is often used by the community and it would be nice to make available both:

● the DA-L63 setup,

● together with the MPC code presented in the paper, to ensure reproducibility of the results and provide a quickstart initial setup for future works and people interested in collaborating on this topic.

Response: Archiving the source code is underway and will be opened on GitHub (https://github.com/) prior to the final publication.

**[Reviewer 2 Technical corrections]**

(1) l.20-22: The authors used exactly the same 2 first sentences for the abstract and Introduction. This has to be modified.
Response: Thanks for your careful reading. We revised this point (L10).

(2) l.26: In the sentence "Under the program, researchers are exploring various engineering manipulations such as cloud-seeding and atmospheric heating," the word "manipulations" may be replaced with "techniques" for a more precise and formal tone. Or at least, well define what you intend by "manipulations".
Response: Thanks for your careful reading. We revised this point (L26).

(3) l.127: The word "solve" in the sentence "Thus, this study solve them using a numerical approach" should be changed to "solves.
Response: Thanks for your careful reading. We revised this point (L131).

(4) l.128: In the sentence "Given the first guess of control inputs $\mathbf{u}^*(\tau; t)$, temporally forward computations (Eqs. 7 and 8) are performed to obtain $\mathbf{x}^*(\tau; t)$ from $\tau = 0$ to $\tau = T$," the term "temporally" should be changed to "temporal" for accuracy.
Response: Thanks for your careful reading. We revised this point (L132, L135, and L378).

(5) l. 366: The phrase "the constraint impose more difficulty" should be corrected to "the constraint imposes more difficulty."
Response: Thanks for your careful reading. We revised this point (L333).

(6) l.326: In the sentence "In addition, the MTC decrease for smaller $U$," the verb should be in singular form as "decreases" to match the subject "MTC." In the same sentence, "but the SR and MTF worsen accordingly" could be improved by adding "do" before "worsen" for better clarity.
Response: Thanks for your careful reading. We revised this point (L334).